# Hemp (*Cannabis sativa* L.) Interruption Cultivation Evidently Decreases the Anthracnose in the Succeeding Crop Chilli (*Capsicum annuum* L.)

Feihu Liu [1], Xuan Li [1,†], Huaran Hu [2,†], Jiaonan Li [1], Guanghui Du [1], Yang Yang [1], Kailei Tang [1,*], E Li [1], Huiping Li [3] and Li Chen [3]

[1] Laboratory of Plant Improvement and Utilization, Yunnan University, Chenggong District, Kunming 650500, China; liufh@ynu.edu.cn (F.L.)

[2] Cash Crops Research Institute, Yunnan Academy of Agricultureal Sciences, Kunming 650221, China

[3] Yanshan Station of Cash Crops, Yanshan 663100, China

**\*** Correspondence: author: kailei.tang@ynu.edu.cn; Tel.: +86-13629441910

**†** These authors contributed equally to this work.

**Abstract:** Continuous cropping increases disease severity and causes arrested development of chilli plants and the decrease of yield and quality. Hemp-chilli rotation cropping evidently eased chilli diseases, but the causation remains unknown. This paper investigated the disease index (DI) of chilli's anthracnose for hemp-chilli and continuous chilli cropping, the antifungal effect of water extract of hemp and chilli residues, and bacteria antagonistic to chilli anthracnose fungus from hemp and chilli rhizosphere. Hemp grown as a preceding crop decreased anthracnose DI from 35–39% for the continuous chilli to 14–15% for hemp-chilli rotation. Hemp residue water extract executed suppression of chilli anthracnose fungus and the efficiency increased as the extract concentration increased from 1% to 5%. Hemp extract concentration 5% gave a mean inhibition ratio (IR) of 32.34% to spore germination and IR 53.72% to mycelia growth, which was much greater than that of the chilli extract. Antagonistic bacteria isolated from the hemp rhizosphere evidently depressed the mycelia growth of the fungus with a mean IR 32.35%, while no antagonistic bacteria were obtained from the chilli rhizosphere. The stronger allelopathy of preceding hemp plants and antagonistic bacteria from the hemp rhizosphere synergistically suppressed the fungus growth and eased the disease in the succeeding chilli crop.

**Keywords:** chilli (*Capsicum annuum* L.) anthracnose; hemp (*Cannabis sativa* L.); allelopathy; antagonistic bacteria

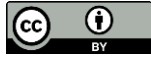

## 1. Introduction

Chilli (*Capsicum annuum* L.) is a kind of important vegetable and condiment, widely grown in different countries, although China is the largest producer in the world [1]. Because of the limitation of farm lands, soil and climate adaptability, etc. [2,3], chilli continuous cropping is hardly avoided; although chilli is a crop suffering from severe continuous cropping obstacles mainly resulting from soil borne diseases. Of the important diseases of chilli, the anthracnose, caused by fungi *Colletotrichum gloeosporioides* (Penz.) Sacc. Or/and *C. capsici* (Syd.) Butl. or/and *C. phomoides* (Sacc.) Chest., is one of the main soil borne diseases [4,5]. This disease causes a large amount of defoliation and fruit rot, even the death of chilli plants during the later growth period, leading to a 20—30% decrease in fruit yield and much greater decrease in quality. Continuous cropping obstacles have also been widely found on other crops, showing growth retardation, with a decrease of yield and product quality, mainly caused by the aggravation of diseases [6–8]. The development of biological controlling for plant diseases has been focused for a long time owing

to its effectiveness and sustainability. In fact, rotation cropping is an important scheme of biological controlling that can effectively alleviate plant diseases so as to ease continuous cropping obstacles of crops [6,9–11], although some researchers have used microbes as highly effective biological controlling agents for plant (such as tomato) disease [12].

Hemp (*Cannabis sativa* L.), one of the crops utilized earliest by humans, is an annual plant of the genus *Cannabis* in family Cannabaceae cultured worldwide owing to its very good adaptability to diversiform eco-environments [13]. Industrial hemp is defined as the cultivar type of *Cannabis* plant that has the psychotoxic chemical $\Delta^9$-tetrahydrocannabinol less than 0.3% (dry weight) in any parts of the plant. Some experiments testified that hemp cultivation increased the yield and eased the diseases of the succeeding crops. For example, hemp cultivation increased the yields of succeeding wheat in the following first and second years compared to the continuous cropping wheat [14]. In the soil of continuous planting soybean for seven years, soybean after hemp decreased the number of second-stage juveniles of soybean cyst nematode (*Heterodera glycines* Ich.) by 29.8% over the continuous soybean. The number of soil-borne pathogens of *Fusarium*, *Rhizoctonia*, and *Pythium* after hemp planting and root rot disease index were remarkably lower than the continuous soybean [15]. Moreover, scientists revealed that the allelopathic effect of hemp accelerated the germination of melon broomrape (*Phelipanche aegyptiaca* Pers.) and sunflower broomrape (*Orobanche cumana* Wallr.) seeds [16].

Quite many farmers reported that in the fields of continuously planting cruciferous or solanaceous vegetables, vegetables after hemp (hemp-vegetable rotation) evidently eased the incidence of diseases. These cases were further confirmed by our investigations from different districts (data not published), although the knowledge at that time was not very precise and the reasons were far from being explored. To this end, we chose the important chilli disease anthracnose and carried out the study, including field investigation and laboratory tests, aimed at clearly verifying the reduction of chilli anthracnose because of hemp interruption cultivation (hemp-chilli rotation cropping), further exploring why the hemp interruption cultivation could reduce the incidence of anthracnose of the succeeding crop chilli.

In this paper, we firstly reported the reduction of chilli's anthracnose after hemp planting in comparison with the continuous chilli field, and further tested the antifungal effect of water extract of hemp and chilli residues against chilli anthracnose fungus. We further isolated the potential antagonistic bacteria of chilli anthracnose fungus from the hemp and chilli rhizosphere, in order to partially reveal the biological reasons for hemp cultivation suppressing diseases of succeeding crop chilli. Thus, we provide a theoretical understanding for overcoming the continuous cropping obstacles of vegetables such as chilli by means of hemp interrupt planting.

## 2. Materials and Methods

### 2.1. Investigation of Chilli Anthracnose in the Fields of Continuous Chilli and Hemp-Chilli Rotation

The field investigation was performed in Yanshan County (major provincial production area of chilli), south-east of Yunnan Province, China. The experimental field (chilli was planted in the previous year) was randomly divided into rotation cropping (of hemp-chilli) and continuous cropping (of chilli) plots (50 m²/plot) with four replicates. Hemp (cultivar Yunma 1) was planted in the rotation plots and chilli (cultivar 'red sun' and local race 'ram horn', in the 25 m² subplot, with the two chilli cultivars randomly arranged) in the continuous plots in the first year. Chilli then was planted in all the plots in the second year, with the two chilli cultivars randomly arranged. The field experiment was carried out at three separate sites to insure the reliability of field disease investigation. The anthracnose disease of chilli was investigated by sampling randomly 100 fruits from each plot per chilli cultivar in both the rotation and continuous cropping regimes at the harvest

stage. The lesion area and total surface of the fruit were scanned using a YMJ-CHA3 intelligent leaf area system (accuracy ±2%) provided by Zhejiang Topu Yunnong Technology Co., Ltd. China. For the convenience of calculation, the disease severity was classified according to the percentage of lesion area/total surface of the fruit into six grades, i.e., 0 (lesion free), 1 (<2%), 3 (2–8%), 5 (9–15%), 7 (16–25%), and 9 (>25%). The disease index (DI) of anthracnose and controlling effect (CE) were worked out as DI (%) = Σ (number of diseased fruits' corresponding grade)/(total number of fruits examined' the highest grade 9) × 100; CE (%) = (DI of continuous cropping−DI of rotation cropping)/DI of continuous cropping × 100.

### 2.2. *Suppression of the Growth of Chilli Anthracnose Fungus by Plant Residue Extract*

The experiment was carried out at Yunnan University, Kunming, China using a completely random design. The industrial hemp cultivar 'Yunma 1' and the isolated chilli anthracnose fungus (*Colletotrichum capsici* (syd.) Butl.) were kindly provided by Dr Yang Ming and Ms Chen Jing, Yunnan Academy of Agricultural Sciences.

### 2.2.1. Preparation of Plant Residue Water Extract

Hemp and chilli (local race 'ram horn') seeds were sown in pots containing sterilized field soil (collected from the experimental sites, steam sterilization for 60 min) and cultured in a glass house, and the plants were taken out with roots intact at the stage of twelve leaves. Soil was washed off from the roots carefully. The plants were parted into leaf, stem, and root, inactivated at 105 °C for 30 min, and dried at 80 °C in an oven. According to the method suggested by [17], the pulverized samples of leaf, stem, and root were separately extracted in conical flasks using distilled water in a ratio of plant powder/water (*w/v*) of 1:20. The flasks were shaken at 160 rpm and 30 °C for 24 h, and the extracts were filtered and then stored at 4 °C.

### 2.2.2. Preparation of Spores' Suspension of Chilli Anthracnose Fungus

A bit of pre-cultured chilli anthracnose fungus was inoculated to the center of PDA (potato dextrose agar) medium (in Petri dishes) using a needle and cultured under conditions of 28 °C constant temperature in the dark for seven days until the colony covered the dish. The mycelia were transferred into a flask containing 100 mL sterilized water and were shaken well. The solutions were filtered using four layers of gauze and the filtrate (spores suspension) was adjusted into $10^3$ spores/mL using a hemocytometer under an electronic microscope [18].

### 2.2.3. Suppression of the Growth of Chilli Anthracnose Fungus by Plant Residue Extract

Inhibition of Spore Germination

This test was performed following the method recommended by [19] with three replicates. PDA medium was sterilized and cooled down to 55 °C. Hemp (or chilli) residue extract was then added into the medium with the ratios of extract/medium (*v/v*) 0 (control), 1%, 3%, and 5%. The mixed medium solution was poured into Petri dishes and 100 μL spore suspension of the anthracnose fungus was spread on the medium when the medium cooled down to solid state. Sterilized water was used as control. The Petri dishes were cultured under 28 °C constant temperature in the dark for two days and the colonies (each represented a germinated spore) were counted. The inhibition ratio of spore germination ($IR_{SG}$) was worked out using the formula: $IR_{SG}$ (%) = (number of spores germinated for the control−number of spores germinated for the treatment)/number of spores germinated for the control × 100.

Inhibition of Mycelia Growth

Water-agar medium (1%, *w/v*) was used to conduct this test following the method suggested by [20] and [18] with three replicates. The medium was sterilized and cooled down to 55 °C. Hemp (or chilli) residue extract, sterilized by filtration, was added into the medium with the ratios of extract/medium (*v/v*) 0 (control), 1%, 3%, and 5%. The mixed medium was shaken well and poured into Petri dishes, 20 mL for each. The medium in the dish was blown dry and inoculated with the fungus mycelia (a tiny colony cultured from spores suspension) at the center of the medium on a sterile bench. The dishes were cultured at 28 °C constant temperature in the dark for four days. Then, the diameters of hyphae were measured using a Motic BA310 biological microscope and Motic Images plus 3.0 software, and diameters of colonies were tested using cross measurement. The inhibition ratio of mycelia growth ($IR_{MG}$) was calculated based on the suppression of plant residue extract to both the hypha diameter and colony diameter as follows: $IR_{MG}$ (%) = [((hypha diameter of the control—hypha diameter of the treatment)/hypha diameter of the control) + ((colony diameter of the control—colony diameter of the treatment)/colony diameter of the control)]/2 × 100.

### 2.3. Isolation and Identification of Potential Antagonistic Bacteria from Plant Rhizosphere

2.3.1. Preparation of Rhizosphere Soil Suspension

The dilution plating method was applied to prepare the soil suspension [18,21]. Hemp or chilli (local race 'ram horn') seeds were sown in pots containing sterilized field soil (collected from the experimental sites) and cultured in a glass house. The rhizosphere soil was collected and dried in the air when the seedlings grew to the stage of twelve leaves. Five grams of air-dried soil, 45 mL sterilized water, and two sterile glass balls were put into a flask and shaken for 20 min at 170 rpm, then left to rest for 15 min. A 100-μL suspension taken from the flask was added to a 1.5-mL centrifuge tube containing 900 μL sterilized water and mixed well, to prepare the soil suspension with the concentration $10^{-1}$. The soil suspension was then further diluted to the concentrations $10^{-2}$, $10^{-3}$, $10^{-4}$, and $10^{-5}$, and stored at 4 °C.

2.3.2. Culture of Potential Antagonistic Bacteria from the Soil Suspension

One hundred μL of soil suspension at different concentrations ($10^{-1}$, $10^{-2}$, $10^{-3}$, $10^{-4}$ or $10^{-5}$) was collected and well spread on the surface of LB medium plates using a flame-sterilized glass rod. The plates were sealed up with sealing film, cultured upside down under 30 °C constant temperature in the dark for 48 – 72 h, and stored at 4 °C [18]. The experiments were replicated three times for each suspension concentration, using sterilized water as control.

2.3.3. Screening of the Antagonistic Bacteria

Preparation of Chilli Anthracnose Fungus

A bit of pre-cultured chilli anthracnose fungus (mycelia) was inoculated to the center of PDA medium (in Petri dishes) using a needle and cultured under conditions of 28 °C constant temperature in the dark for four days until the colony covered the semidiameter of the dish.

Test of Antagonistic Bacteria

The modified plating confrontation culture method suggested by [22] was applied to test the antagonistic bacteria. A puncher of 5 mm diameter was used to punch at the center of the prepared PDA medium (in plates). The activated anthracnose fungus mycelia were then inoculated in the punch. The plates were culture upside down under 28 °C constant temperature in the dark for 24 h. Then, 200 colonies (potential antagonistic bacteria) were selected using a sterilized toothpick from each plate prepared as in Section 2.3.2. Four of

them were inoculated with equal distance (2 cm apart) lining on the diagonal in each plate surrounding the anthracnose fungus (in the punch) and cultured under 30 °C constant temperature in the dark for two days. The antagonistic bacteria were selected according to the diameters of bacteriostatic rings. This experiment was performed with three replicates, using the potential antagonistic bacteria free plates as control.

Re-Test of the Antagonistic Bacteria

The test of antagonistic bacteria was performed once more using the selected antagonistic bacteria following the above protocol, and the strongest antagonistic bacteria were screened out.

### 2.3.4. Suppression of the Growth of Chilli Anthracnose Fungus by the Selected Antagonistic Bacteria

The suppression effect of the selected bacteria on the growth of anthracnose fungus was tested following the protocol in Section 2.3.3 using un-inoculated plates as control. The diameters of anthracnose fungus colonies were measured using a cross method and the inhibition ratio of fungus in colony diameter ($IR_{CD}$) was worked out as follows: $IR_{CD}$ (%) = (colony diameter of the control—colony diameter of the treatment)/colony diameter of the control × 100.

### 2.3.5. Identification of the Antagonistic Bacteria Based on 16SrDNA Sequence

The 16SrDNA was amplified using the polymerase chain reaction protocol, with total DNA isolated from the culture of antagonistic bacteria as template and the primers 27F5′-AGAGTTTGATCMTGGCTCAG-3′/1492R5′-TACGGYTACCTTGTTACGACTT-3′. The PCR products were checked using 1% agarose gel electrophoresis and sent to the company TsingKe Biological Technology Beijing for 16SrDNA sequencing (primers were the same as above). The sequences were registered in the GenBank database and multiple sequence alignments were finished using the website EZBioCloud [23]. The phylogenetic analysis was performed using the MEGA software package (ver 11) to construct a phylogenetic tree based on the neighbor-joining method [24].

### *2.4. Data Collection and Analysis*

All the tests and investigations were replicated at least three times. The data for disease indexes of chilli anthracnose in the fields, inhibition of hemp or chilli residue extract to spore germination and fungus growth, as well as data for inhibition of chilli anthracnose fungus caused by the isolated antagonistic bacteria, were subjected to statistical analysis using Data Processing System (DPS) software v19.05 [25]. The differences in treatments were determined using LSD test.

## 3. Results

### *3.1. Effect of Hemp Cultivation on the Incidence of Anthracnose Disease of Succeeding Chilli*

The investigation results in the field showed the anthracnose disease index of the rotation chilli (chilli after hemp) was much lower than that of the continuous chilli. The former was 14–15% and the latter was 35–39%, presenting a more than 60% controlling effect of hemp cultivation to anthracnose of the succeeding crop chilli, although the incidence of disease in between cultivars was not varied evidently (Figure 1).

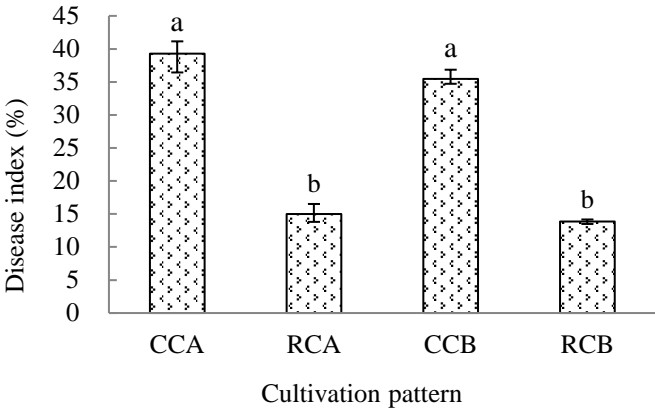

**Figure 1.** Disease indexes of chilli anthracnose in the fields of hemp-chilli rotation and chilli continuous cropping collected from three experimental sites. CCA, continuous cropping of chilli "ram horn"; CCB, continuous cropping of chilli "red sun"; RCA, rotation cropping of hemp-chilli "ram horn"; RCB, rotation cropping of hemp-chilli "red sun". Different letters on the top of bars show significant difference at 0.05 levels tested by LSD.

*3.2. Effect of Hemp and Chilli Extract on Spore Germination of the Anthracnose Fungus*

Within the set concentrations, 1%, 3%, and 5%, the inhibition effect to spore germination of the anthracnose fungus increased as the increase of the concentration of plant residue water extract, so that the number of germinated spores decreased correspondingly (data not shown). The extracts from different plant parts, including leaf, stem, and root all showed definite suppression of spore germination of the fungus, but the suppression effect showed the order as root > stem > leaf, and the mean inhibition ratio (IR) of root extract was remarkably greater than that of the leaf extract (Figure 2). For the hemp residue extracts, the 5% extract gave the strongest effect of IR 31.34% on average, including root, stem, and leaf, while 5% root extract imposed IR 36.57% to spore germination of the anthracnose fungus. Moreover, the inhibition effect of hemp extract on the spore germination of anthracnose fugus was much greater than that of chilli extract (Figure 2).

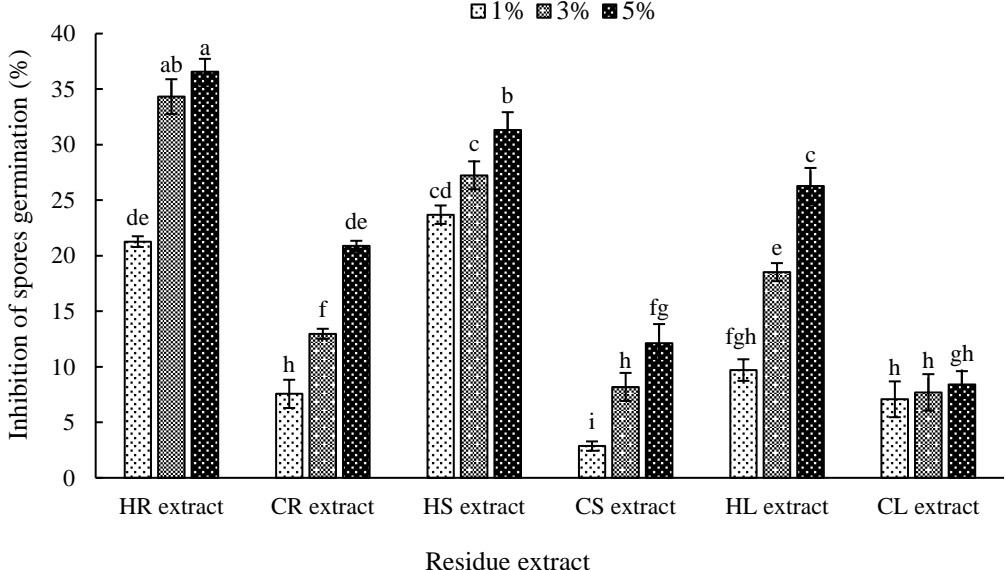

**Figure 2.** Inhibition of hemp or chilli residue extract to spore germination of chilli anthracnose fungus. HR, hemp root; CR, chilli root; HS, hemp stem; CS, chilli stem; HL, hemp leaf; CL, chilli leaf. Different letters on the top of bars show significant difference at 0.05 levels tested by LSD.

*3.3. Effect of Hemp and Chilli Extract on Mycelia Growth of the Anthracnose Fungus*

The suppression of plant residue water extract to the growth of the anthracnose fungus also presented concentration effects, as with the increase of the extract concentration, the diameters of fungus hyphae and colonies reduced accordingly (Figures 3 and 4). Hemp extracts of different plant parts (leaf, stem, and root) all showed significant inhibition to the growth (diameter) of fungus hyphae, although the inhibition effects of the extracts from different plant parts varied little, not showing definite organ (leaf, stem, and root) differences (Figure 3). On the other hand, it was determined that hemp extracts from leaf, stem, or root all reduced evidently the diameters of anthracnose fungus colonies (Figure 4). Of the extracts of different concentrations from different plant parts of hemp, 5% leaf extract showed the strongest inhibition to the growth of the fungus, giving the minimum number of colonies that almost was one seventh of the control (0%) in colony diameter (Figure 4). Combining the suppression of hemp extract with the diameters of hyphae and colonies of the anthracnose fungus, the comprehensive IR was as high as 53.72%, while the best inhibition effect (IR 62.52%) to fungus growth was obtained using the 5% leaf extract that was evidently better than other treatments, including extracts from other plant parts (root, stem) or concentrations (1%, 3%) (Figure 5). Again, it was observed that chilli extract showed evidently less suppression of fungus growth than hemp extract, although chilli extract also inhibited fugus growth to some extent (Figures 3–5).

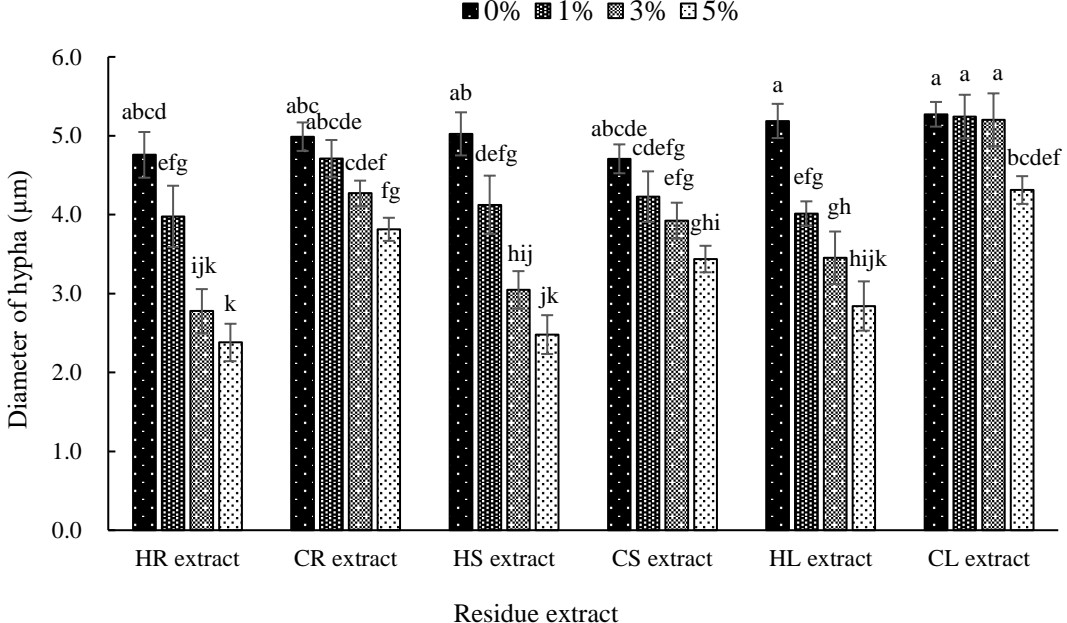

**Figure 3.** Decrease in hyphae diameter of chilli anthracnose fungus caused by hemp or chilli residue extract. HR, hemp root; CR, chilli root; HS, hemp stem; CS, chilli stem; HL, hemp leaf; CL, chilli leaf. Different letters on the top of bars show significant difference at 0.05 levels tested by LSD.

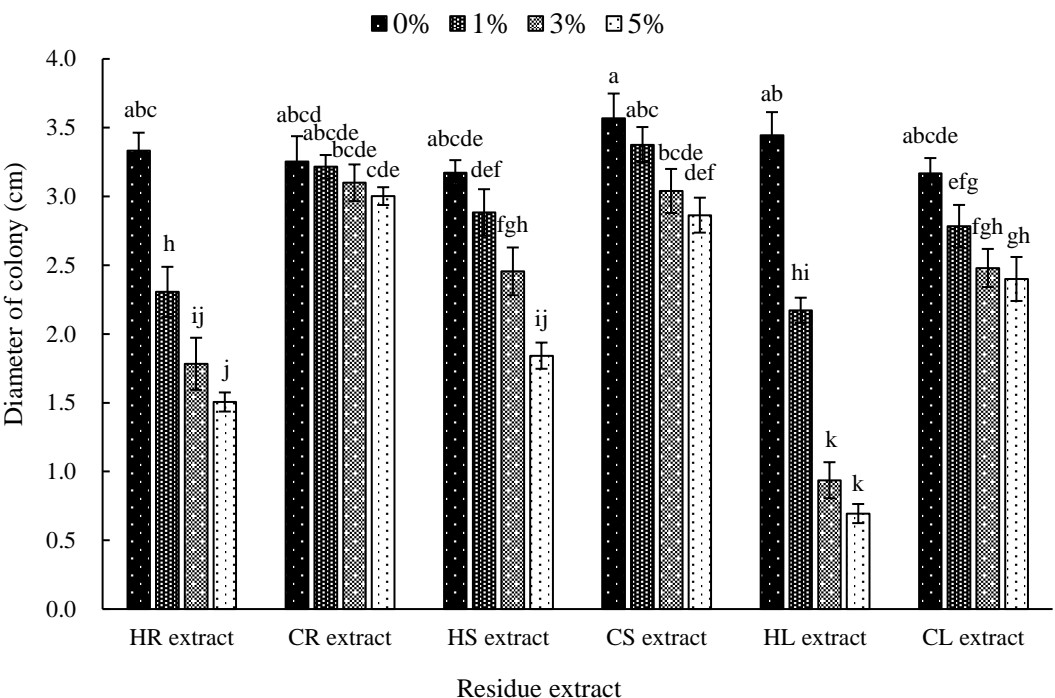

**Figure 4.** Decrease in colony diameter of chilli anthracnose fungus caused by hemp or chilli residue extract. HR, hemp root; CR, chilli root; HS, hemp stem; CS, chilli stem; HL, hemp leaf; CL, chilli leaf. Different letters on the top of bars show significant difference at 0.05 levels tested by LSD.

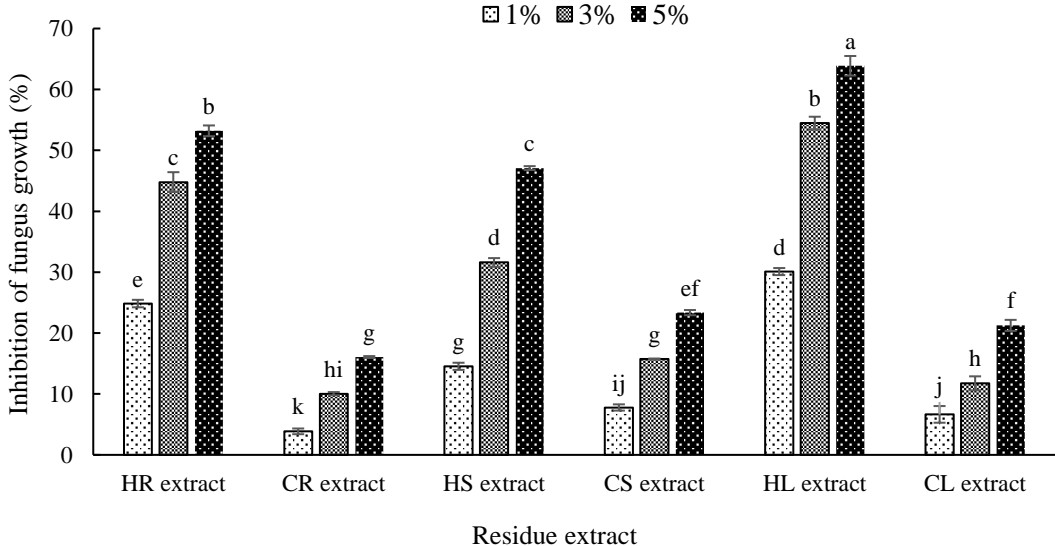

**Figure 5.** Comprehensive inhibition effect of hemp or chilli residue extract to chilli anthracnose fungus based on the diameters of hyphae and colonies. HR, hemp root; CR, chilli root; HS, hemp stem; CS, chilli stem; HL, hemp leaf; CL, chilli leaf. Different letters on the top of bars show significant difference at 0.05 levels tested by LSD.

### 3.4. Antagonistic Bacteria of the Anthracnose Fungus Isolated from Hemp Rhizosphere Soil

#### 3.4.1. The Isolated Antagonistic Bacteria

From the cultures of plant rhizosphere soil suspension on LB medium (Figure 6), 200 strains (colonies) of rhizosphere bacteria (potential antagonistic bacteria) were picked out from the cultures of soil suspension ($10^{-3}$ for hemp and $10^{-2}$ for chilli) and co-cultured with the anthracnose fungus. Of these co-cultures, 15 strains were screened out from hemp rhizosphere soil that were evidently antagonistic to chilli anthracnose fungus, but no bacterium antagonistic to chilli anthracnose fungus was obtained from chilli rhizosphere soil (Figure 7).

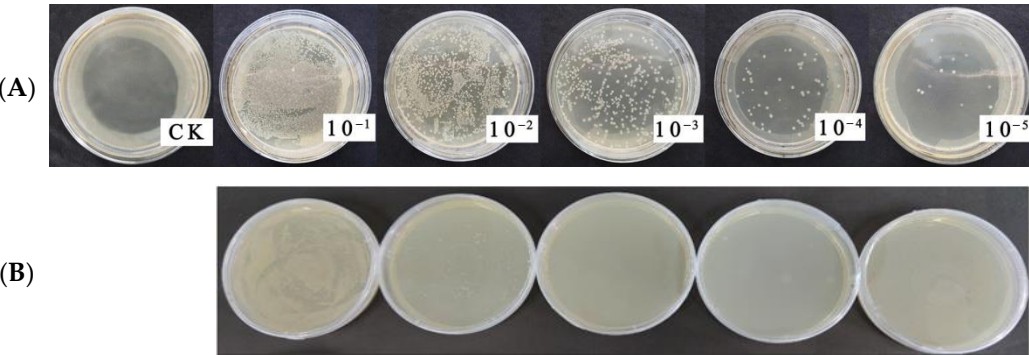

**Figure 6.** Colonies of potential antagonistic bacteria on LB medium cultured from hemp (**A**) or chilli (**B**) rhizosphere soil suspension. CK, the control, soil suspension free; $10^{-1}$ – $10^{-5}$ show the concentrations of soil suspension from hemp or chilli rhizosphere.

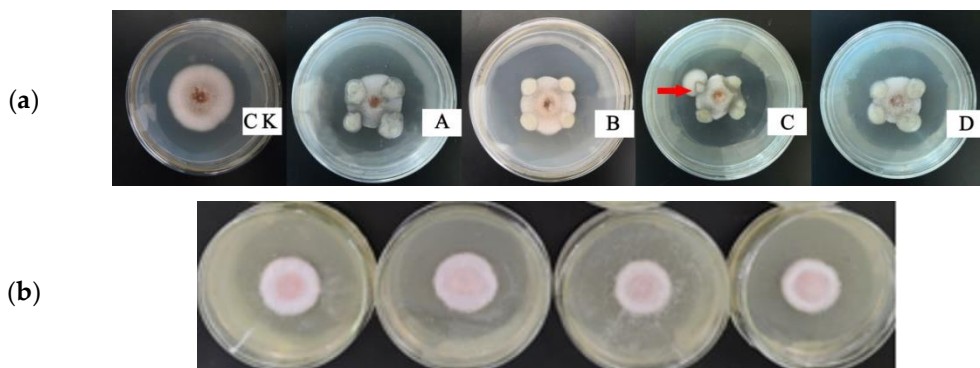

**Figure 7.** Antagonistic bacteria from hemp (**a**) or chilli (**b**) rhizosphere soil re-tested on PDA medium. CK, the control, anthracnose fungus hyphae cultured on PDA; A, B, C and D, anthracnose fungus hyphae cultured on PDA with 4 different strains of suspicious antagonistic bacteria isolated from hemp rhizosphere inoculated in each Petri dish. Exception of the left-up strain marked by an arrow, other 15 strains were determined as antagonistic bacteria. Correspondingly, anthracnose fungus hyphae cultured on PDA with 4 different strains of suspicious antagonistic bacteria isolated from chilli rhizosphere inoculated in each Petri dish.

#### 3.4.2. Identification of the Antagonistic Bacteria Based on DNA Sequence

The particular 16SrDNA was amplified using total DNA from the 15 strains of antagonistic bacteria isolated from hemp rhizosphere soil as template and specific primers (Figure 8). The PCR products were approximately 1400 bp and the sequences were consistent with the data obtained from searching websiteEZBioCloud. The sequences of the isolated antagonistic bacteria were aligned with the closely related subject sequences from EZBioCloud using the website EZBioCloud, showing 99.63–100% similarity with the sequence of related species *Enterobacter sichuanensis* (Table 1). A phylogenetic tree based on the

neighbor-joining method for the isolates along with their closest relatives of genus *Enterobacter* is shown in Figure 9. The 15 strains of antagonistic bacteria were all identified as *Enterobacter* species, belonging to the family Enterobacteriaceae.

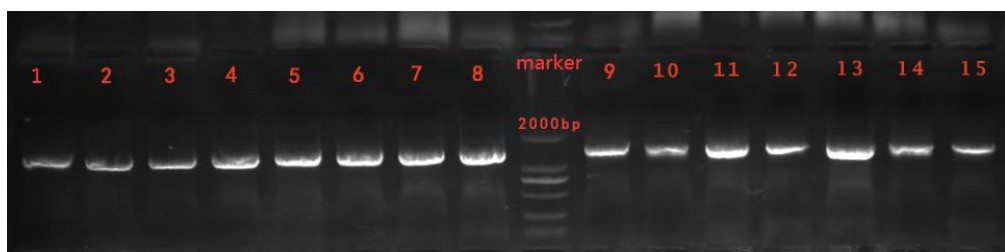

**Figure 8.** PCR products of the 15 strains of antagonistic bacteria. Numbers 1 through to 15 represents the code of isolated antagonistic bacteria, corresponding with Seq 1 through to Seq 15 in Figure 9.

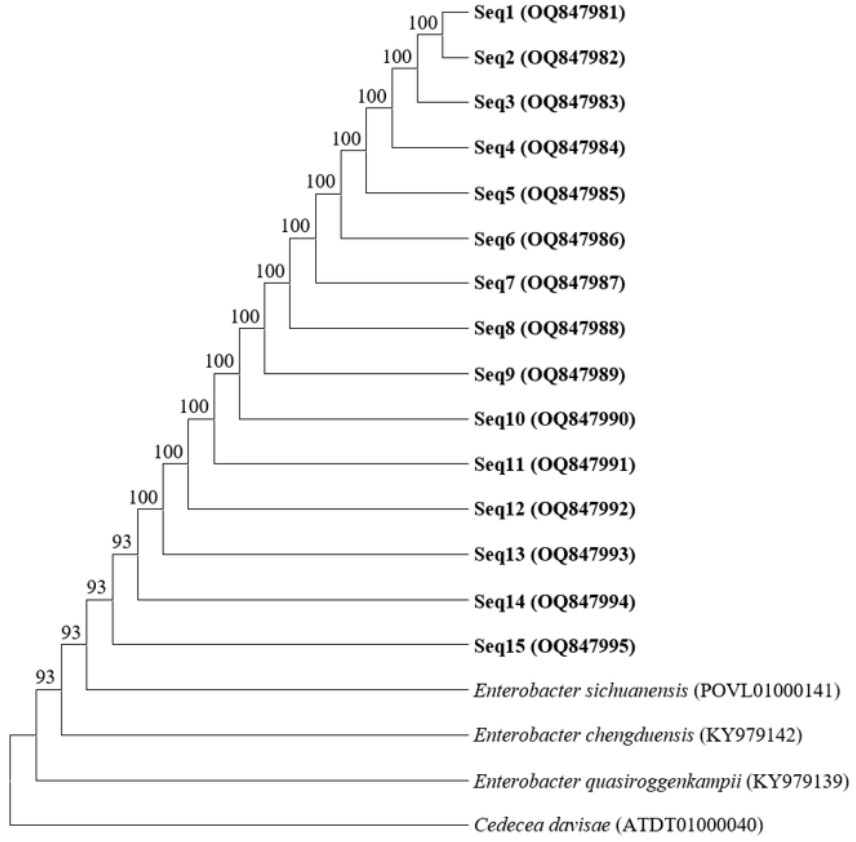

**Figure 9.** Phylogenetic tree for the isolated antagonistic bacteria. Codes in the parentheses are the accession numbers in GenBank.

**Table 1.** The 15 strains of antagonistic bacteria (AB) from hemp rhizosphere soil all belonging to the family Enterobacteriaceae.

| Code of AB | GenBank Accession Number | Taxon of the Strains | Sequence Similarity (%) |
|---|---|---|---|
| 1 | OQ847981 | *Enterobacter sichuanensis* | 99.86 |
| 2 | OQ847982 | *Enterobacter sichuanensis* | 99.74 |
| 3 | OQ847983 | *Enterobacter sichuanensis* | 99.63 |
| 4 | OQ847984 | *Enterobacter sichuanensis* | 99.74 |

| 5 | OQ847985 | *Enterobacter sichuanensis* | 99.74 |
| 6 | OQ847986 | *Enterobacter sichuanensis* | 99.74 |
| 7 | OQ847987 | *Enterobacter sichuanensis* | 100.00 |
| 8 | OQ847988 | *Enterobacter sichuanensis* | 100.00 |
| 9 | OQ847989 | *Enterobacter sichuanensis* | 99.86 |
| 10 | OQ847990 | *Enterobacter sichuanensis* | 100.00 |
| 11 | OQ847991 | *Enterobacter sichuanensis* | 99.86 |
| 12 | OQ847992 | *Enterobacter sichuanensis* | 100.00 |
| 13 | OQ847993 | *Enterobacter sichuanensis* | 100.00 |
| 14 | OQ847994 | *Enterobacter sichuanensis* | 100.00 |
| 15 | OQ847995 | *Enterobacter sichuanensis* | 100.00 |

### 3.4.3. Inhibition of the Isolated Antagonistic Bacteria on Growth of the Anthracnose Fungus

Four different strains of antagonistic bacteria isolated from hemp rhizosphere soil co-cultured with the anthracnose fungus in each plate remarkably suppressed the growth of the fungus (Figure 7). Although the suppression of the growth of the fungus by different strains of antagonistic bacteria varied to some extent, colony diameters of the fungus were significantly decreased due to the isolated strains of antagonistic bacteria compared to the control (Figure 10) with a mean IR (of colony diameter) up to 32.35%.

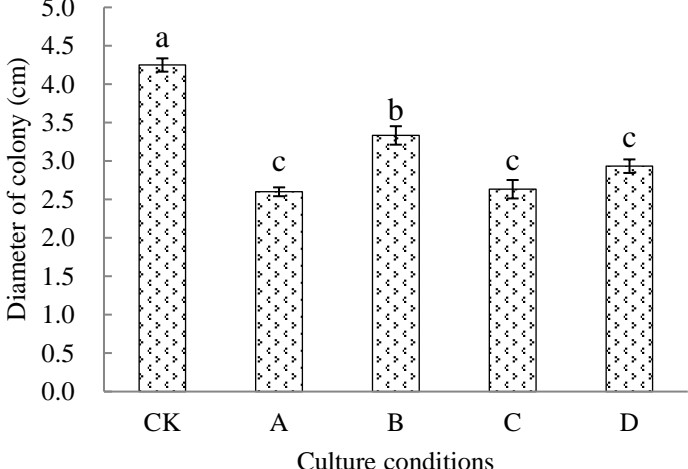

**Figure 10.** Decrease in colony diameter of chilli anthracnose fungus caused by the co-cultured antagonistic bacteria isolated from hemp rhizosphere. Different letters on the top of bars show significant difference at 0.05 levels tested by LSD. CK, the control, anthracnose fungus hyphae cultured on PDA; A, B, C and D, anthracnose fungus hyphae cultured on PDA with different four strains of antagonistic bacteria isolated from hemp rhizosphere inoculated in each Petri dish.

## 4. Discussion

Anthracnose is a severe challenge in chilli production to both the agriculturists and farmers, especially in the continuous cropping system, but unfortunately the efficient management strategy for this disease is not yet available so far. Although the disease could be controlled to some extent by chemical bactericides, their excessive use has already caused popular anxiety concerning the safety of foods and the agricultural eco-system, especially with regard to the sustainability of agricultural development [26] necessary to the survival and development of the human being.

A smart biological approach for controlling crop diseases to decrease or supersede the use of chemicals is certainly a generally accepted scheme to maintain the productivity of crops and support the ecological environment. It is known that allelopathy has a very

important function to help crops in resisting the diseases, and the common allelochemicals are benzoxazinoids, a family of cyclic hydroxamic acids synthesized by a range of plant species [27]. Researchers revealed that wheat allelochemicals were toxic to fungus, which was effective for inhibiting the growth of cereal crop pathogens [28]. Hemp cultivation was also observed to be evidently inhibited by some pathogens of the succeeding crops, such as root-knot nematode of tomato [29] and cyst nematode, or soil-borne pathogens *Fusarium*, *Rhizoctonia*, and *Pythium* of soybean [15]. Potential allelochemicals were also observed be effective to inhibit plant diseases or pathogens, such as *Capsella bursa-pastoris* extracts against *Ralstonia solanacearum* [30], allelochemicals from watermelon and rice roots suppressive to watermelon *Fusarium* wilt [19], and the antibacterial activity of active substances from *Amaranthus tricolor* [17], etc. We found that hemp cultivation remarkably reduced the diseases of the succeeding chilli and presumed that the allelopathy of hemp plants or/and antagonistic bacteria from hemp rhizosphere might be toxic to the pathogens of the chilli. Experiments were therefore carried out to verify the presumption, and positive results were obtained as hemp residue extracts significantly depressed the spore germination and mycelia growth of chilli anthracnose fungus, and the effect was much greater than that of chilli extract.

Some actinomyces, screened out from the plant rhizosphere in a previous study, identified as *Streptomyces violaceoruber*, showed suppression of chilli anthracnose fungus and promotion of plant growth [31–33], as well as the antagonistic bacteria against *Phytophthora nicotianae* isolated from tobacco rhizosphere [34]. In our study, antagonistic bacteria were screened out from the hemp rhizosphere soil showing high efficiency to depress the growth of the anthracnose fungus and were identified into species in genera *Enterobacter*, family Enterobacteriaceae, but none was obtained from chilli rhizosphere. Other researchers also found some specific strains from the family Enterobacteriaceae that could inhibit some bacterial strains. Of these bacterial strains, 34.5% from genus *Enterobacter* and 14% from *Leclercia* showed inhibition to other bacteria [35]. These bacteria exude lipopeptides, such as surfactins, iturins, and fengycins, and perform definitely antagonistic activity for a wide range of potential phytopathogens, including bacteria, fungi, and oomycetes [36,37]. These results implied a promise to develop a bio-bactericide using antagonistic microbes [38].

Summarizing the discussion above, the results in our study indicated that the isolated antagonistic bacteria of the anthracnose fungus, *Enterobacter* species such as *Enterobacter sichuanensis*, might be a candidate for developing a bio-fertilizer and/or bio-control agent for growing chilli resistant to anthracnose, and thus to cope with or ease the continuous obstacles of the crop.

This experiment did not support whether there was a stronger inhibition of spore germination and/or mycelia growth of the anthracnose fungus if the concentration of the extract is greater than 5%, although a higher concentration of plant residue extract in the field is not very possible. So, research work is being carried out focusing on the bacteriostasis of a higher concentration of hemp extract, the chemicals from the extract inhibiting the fungus, and how they work, as well as the biological process (mechanisms) of antagonistic bacteria isolated from the hemp rhizosphere in inhibiting the growth of the anthracnose fungus.

## 5. Conclusions

Hemp planted as a preceding crop imposed a controlling effect of chilli anthracnose of greater than 60%, considering the disease indexes decreased from 35–39% for the continuous chilli to 14–15% for hemp-chilli rotation. Water extract of hemp residue evidently inhibited spore germination and mycelia growth of chilli anthracnose fungus, e.g., 5% hemp root extract expressed IR 36.57% to spore germination, while 5% hemp leaf extract gave IR 62.52% to mycelia growth of the fungus, which were much greater than that of the chilli extract. Antagonistic bacteria were isolated from hemp rhizosphere that presented a mean IR 32.35% to growth of the fungus; they were identified as belonging to the

genus *Enterobacter*, in family Enterobacteriaceae, but no antagonistic bacterium was isolated from the chilli rhizosphere. Therefore, it is reasonable to contemplate that the synergistic reaction of allelochemicals from hemp residues and antagonistic bacteria from hemp rhizosphere soil remarkably mitigated the anthracnose of chilli in the field of hemp-chilli rotation.

**Author Contributions:** Conceptualization, F.L. and K.T.; methodology, F.L., K.T., G.D. and Y.Y.; software, F.L. and J.L.; validation, E.L., H.L. and L.C.; formal analysis, X.L., H.H. and J.L.; investigation, E.L., H.L. and L.C.; data curation, X.L., H.H. and J.L.; writing—original draft preparation, F.L.; writing—review and editing, K.T., G.D. and Y.Y.; supervision, F.L.; project administration, X.L., H.H., J.L., G.D., K.T. and Y.Y.; funding acquisition, F.L. and K.T. All authors have read and agreed to the published version of the manuscript.

**Funding:** This research was funded by China Agriculture Research System of MOF and MARA under grant number CARS-16-E15; and the National Natural Science Foundation of China under grant number 31871671.

**Data Availability Statement:** Data is unavailable due to privacy.

**Acknowledgments:** We express thanks to Yang Ming and Chen Jing for providing the industrial hemp cultivar 'Yunma 1' and the isolated chilli anthracnose fungus.

**Conflicts of Interest:** The authors declare no conflict of interest.

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
