# Peer review of "Hemp (Cannabis sativa L.) Interruption Cultivation Evidently Decreases the Anthracnose in the Succeeding Crop Chilli (Capsicum annuum L.)"

_agronomy, doi:10.3390/agronomy13051228_

Round 1

Reviewer 1 Report

Dear Authors,

in your manuscript “Hemp (Cannabis sativa L.) interruption cultivation evidently decreases the anthracnose in the succeeding crop chilli (Capsicum annuum L.)” you reported the beneficial effects of hemp-chilli rotation cropping in controlling chilli anthracnose respect than the continuous chilli cropping. Moreover, you tested i) the antifungal activity of water extract of hemp and chilli leaves, rhizomes and roots against chilli anthracnose fungus and ii) the antifungal activity of bacterial strains isolated from hemp and chilli rhizosphere against the chilli anthracnose fungus. This study is very interesting and deals with a very important issue such as the environmental sustainability by using biological control of an important chilli pathogenic fungus. In general, in the first part of the manuscript in which is reported the effect of culture rotation and the antifungal activity of culture water extracts no critical issues were found in the methodology and results, except for some minor corrections that you can found directly in the text, while important lack are evident in the methodology and results about bacterial isolation and identification and antifungal assays with the antagonistic bacteria:

Paragraph 2.3.1

Isolation of antagonistic bacteria from hemp and chilli rhizosphere were conducted sowing hemp and chilli seeds in pots containing sterilized soil from the experimental sites. You collected the rhizosphere soil from the pots when the seedling grew to the stage of 12 leaves. Results of this experiment show something unclear:

how it is possible that you didn’t find bacteria from chilli rhizosphere? Many studies report the presence in chilli rhizosphere of several bacteria, also PGPR (e.g. Linu et al., 2019). Moreover, I believe that this experiment should have been carried out on the rhizosphere of experimental fields, in which you registered the inhibition effects of hemp on anthracnose. You could probably have isolated many more antagonistic bacteria, also from chilli rhizosphere. I think that you need to repeat correctly the analysis.

Paragraph 2.3.5

The molecular identification is very superficial. It is not acceptable that you used only BLASTn as tool for identification. You must add a phylogenetic analysis.

Paragraph 3.4.2

None of your sequences was registered on GenBank? Do you have the accession numbers of your 15 bacterial sequences? In table 1 you report the Acc. Numb. of the strains with which your sequence had 100% of similarity. This table is completely useless without the Accession Number of your sequences.

Paragraph 3.4.3

You wrote that 4 bacterial strains had inhibitory effect on anthracnose fungus. What are these 4 strains? Give species and accession number. Moreover, from Figure 7 is hard to determine the antifungal effect of these bacteria. Moreover, about Figure 7, if you tell that you didn’t find antagonistic bacteria from chilli rhizosphere what is the meaning of line B of the figure 7? What is it supposed to represent?

I think that the article needs you correct all the second part of your manuscript (from the isolation of antagonistic bacteria to their antifungal activity) on the basis of my suggestions. Other minor modification are reported in the pdf file.

Best regards

Reviewer 2 Report

The suggestions are present as comments to the attached pdf.

Reviewer 3 Report

The manuscript “Hemp (Cannabis sativa L.) interruption cultivation evidently decreases the anthracnose in the succeeding crop chilli (Capsicum annuum L.)” presents relevant information and can be considered for publication in AGRONOMY journal with major revision.

  Authors are encouraged to review the manuscript following instructions:

The authors should write out in full the genus and species only at the first mention of an organism in the manuscript (eg. Line 107, in table 1, etc). It is advisable to consult specific databases to check the current name with the relative classifiers (e.g., Colletotrichum capsici (Syd. & P. Syd.) E.J. Butler & Bisby > species Fungorum current name: Colletotrichum truncatum (Schwein.) Andrus & W.D. Moore (e.g., Line 40, 305, etc)

indicate sp. or spp. after fungi genera (Line 62, etc);

Fungi classifiers should not be italicized (Line 65, 66 etc);

Citation of Dr Yang and Mr. Jing's collaboration should be indicated in the “Acknowledgments” section (Line 108) “Acknowledgments: In this section you can acknowledge any support given which is not covered by the author contribution or funding sections. This may include administrative and technical support, or donations in kind (e.g., materials used for experiments).” (from https://www.mdpi.com/journal/agronomy/instructions#preparation)

Indicate the sterilization time (cycle) (Line 111);

Replace “PDA” with “Potato dextrose Agar (PDA)” (Line 121);

Replace “ml” with “mL” (Line 124, 146, 162, etc);

Sostituire "μl" con "μL" (riga 133, 163, 164, ecc.);

Sostituire "petri" con "Petri" (linee 135, 178, ecc.);

Non iniziare una frase con un numero (riga 163, 169, ecc.)

Inserire il paragrafo dell'analisi statistica del risultati! "Gli autori hanno bisogno di una sottosezione in i materiali e i metodi sulla progettazione sperimentale e la descrizione statistica. Gli autori dovrebbero riportare dettagli sufficienti del disegno sperimentale e analisi statistiche in modo che un ricercatore indipendente possa riprodurre il loro ricerca. Le informazioni pertinenti potrebbero includere, ma non sono limitate a: disegno sperimentale, fattori e livelli, elenco di termini fissi e casuali (con giustificazione), numero di repliche con unità sperimentali chiaramente identificato, struttura di correlazione per misure ripetute, nonché software con la versione e le procedure o i pacchetti utilizzati." (da https://www.mdpi.com/journal/agronomy/instructions#preparation)

I risultati della ricerca mostrano l'attività antimicrobica del batteri isolati contro agenti patogeni; pertanto sono candidati per agenti di biocontrollo e NON BIOFERTILIZZANTI. (Linea 319-320)

Il titolo e/o la didascalia delle cifre devono riportare il test statistico utilizzato (oltre al valore P) (Fig. 1, 2, 3, 4, 5, 9).

indicare unità di misura (Linea 483)

Figura 8 non è menzionato nel testo. Dove dovrebbe essere aggiunto?

Round 2

Reviewer 1 Report

Dear Authors,

Thanks for making the requested corrections. The manuscript is now ready to be published.

Reviewer 3 Report

Accept Submission - The revised manuscript " Hemp (Cannabis sativa L.) interruption cultivation evidently decreases the anthracnose in the succeeding crop chilli (Capsicum annuum L.)" is suitable for publication in its present form.